# Using a multi-strain infectious disease model with physical information neural networks to study the time dependence of SARS-CoV-2 variants of concern

**Wenxuan Li**[1][☯]**, Xu Chen**[2][☯]**, Suli Liu** [1]*****, Chiyu Zhang**[1]**, Guyue Liu**[1]

**1** School of Mathematics, Jilin University, Changchun, Jilin, China, **2** School of Artificial Intelligence, Jilin University, Changchun, Jilin, China

☯ These authors contributed equally to this work.

\* liusuli@jlu.edu.cn

## Abstract

With the ongoing evolution of severe acute respiratory syndrome coronavirus 2 (SARS-CoV-2) and its increasing adaptation to humans, several variants of concern (VOCs) and variants of interest (VOIs) have been identified since late 2020. These include Alpha, Beta, Gamma, Delta, Omicron parent lineage, and other variants. These variants may show distinct levels of virulence, antigenicity, and infectivity, which require specific defense and control measures. In this study, we propose an $SI_1 \ldots I_n R$ infectious disease model to simulate the spread of SARS-CoV-2 variants among the human population. We combine the proposed epidemic model and reported infected data of variants with physical information neural networks (PINNs) to develop a novel mechanism called VOCs-informed neural network (VOCs-INN). In our experiments, we found that this algorithm can accurately fit the reported data of the British Columbia (BC) province and its five internal health agencies in Canada. Furthermore, it can simulate observed or unobserved dynamics, infer time-dependent parameters, and enable short-term predictions. The experimental results also reveal variations in the intensity of control strategies implemented across these regions. VOCs-INN performs well in fitting and forecasting when analyzing long-term or multi-wave data.

## Author summary

Epidemiologists and mathematicians use epidemic models expressed by parameterized differential equations to describe the complex dynamics of COVID-19 transmission. The model's parameters are possibly time-dependent and adjusted to fit the observed data. However, traditional parameter estimation methods usually assume the parameters are

---

**Data availability statement:** The epidemiological data of BC province and its five internal regions in the main text can be obtained

through the following link: https://bccdc.shinyapps.io/genomic_surveillance/ The epidemiological data of COVID-19 in Italy under discussion available at the following repository: https://github.com/CSSEGISandData/COVID-19. The source code in Python is available at the following repository: https://github.com/jluorganization/VOCs-INN.

**Funding:** S.L. was supported by the National Natural Science Foundation of China (Grant Number: 12301627), the Science and Technology Research Projects of the Education Office of Jilin Province, China (Grant Numbers: JJKH20250046KJ, JJKH20241240KJ), and the Technology Development Program of Jilin Province, China (Grant Number: 20210508024RQ). The funders had no role in study design, data collection and analysis, decision to publish, or preparation of the manuscript.

**Competing interests:** The authors have declared that no competing interests exist.

constant functions or piecewise constant functions, making it challenging to capture the time evolution of SARS-CoV-2 variants transmission among the population.

To address this issue, we propose an $SI_1 \ldots I_n R$ model to simulate the spread of SARS-CoV-2 variants among the human population, then use VOCs-INN to identify time-dependent parameters and learn unobserved dynamics both represented by neural networks. Even with limited observed data, we reconstruct the full-spectrum dynamics and forecast the pandemic of two VOIs in the BC province and its five internal health agencies. Our research findings reveal an interesting aspect of disease transmission dynamics, namely the diversity of control measures in different regions, which is reflected in the different shapes and values of removed rates in different areas within the same region. We also successfully reproduced dynamic features that were not observed in two variants of the virus, including their competition and coexistence relationships.

In addition, we conducted two interesting comparison experiments in the discussion section. One is the comparison of the proposed VOCs-INN method's fitting and prediction performance with the nonlinear least squares on the daily confirmed cases from Italy. Another experiment explores the impact of physical information on the performance of the proposed VOCs-INN method, focusing on three key aspects: the influence of weights, contraction factors, and time interpolation points. Furthermore, the adaptability of the VOCs-INN framework enables its customization to diverse epidemiological scenarios and datasets, furnishing a robust instrument to tackle the intricacies and uncertainties inherent in epidemic challenges.

## Introduction

The evolution and mutation of viruses are typical processes that allow them to adjust to various environments and hosts. SARS-CoV-2, the virus that causes COVID-19, continues to evolve and mutate. Since October 2020, several SARS-CoV-2 variants with significant mutations have emerged and been classified by the WHO as VOCs (variants of concern) or VOIs (variants of interest). These variants exhibit differences in transmissibility, disease severity, immune evasion capabilities, and responsiveness to treatments. According to the WHO's updated definition in March 2023, compared to VOIs, VOCs represent major evolutionary advancements and necessitate substantial public health measures. To evaluate the risk, simulating the transmission mechanism within the population and determining the time-dependent characteristics of SARS-CoV-2 variants is crucial.

Epidemiological models, represented by parameterized differential equations, are essential tools for studying the patterns of COVID-19 transmission and developing prevention and control strategies [1–7]. The input parameters, which may vary over time, can impact the model's predictability. Various methodologies can be used to fit the trend of disease progression and determine the values or ranges of parameters. These methods primarily depend on mathematical models and statistical principles, along with the analysis of actual epidemiological data, such as the least squares method [8], the Markov Chain Monte Carlo (MCMC) approach [9], and the iterative filtering method [10].

The methods mentioned above may face some limitations in practical applications. Furthermore, factors like intervention measures, human activities, and virus mutations could potentially alter the parameters of the epidemic model over time. However, most parameter estimation techniques assume these parameters remain constant over time. Therefore, it can be difficult to accurately estimate how these parameters change over time and capture

the complex dynamics of infectious diseases in real-world situations using these methods. Recently, deep neural networks (DNNs) [11] have emerged as a strong approach for tackling complex problems in various fields. They have been utilized not only in areas like autonomous driving [12], image recognition [13], and natural language processing [14], but also in predicting and detecting the spread of epidemics.

During the COVID-19 pandemic, deep neural networks have proven to be useful in developing various simulation frameworks for predicting the transmission dynamics of the epidemic [15,16]. In these studies, Arora et al. employed different versions of Long Short-Term Memory (LSTM) within Recurrent Neural Networks (RNN) to estimate the number of infected cases in India [17]. Shahid et al. compared predictive models such as Autoregressive Integrated Moving Average (ARIMA), Support Vector Regression (SVR), LSTM, and Bi-directional LSTM (Bi-LSTM) to forecast confirmed cases, deaths, and recoveries in ten major countries affected by COVID-19 [18]. Similarly, Zhou et al. assessed predictive models including LSTM, Bi-LSTM, Generalized Regression Neural Network (GRU), and Dense-LSTM for predicting the trend of confirmed cases, deaths, and recoveries in the time series data of twelve major countries impacted by COVID-19 [19]. Other research has also adopted similar methodologies [20–22]. These studies demonstrate the varied applications of neural network architectures in epidemiological forecasting during the global health crisis. Although these models exhibit good data fitting and short-term prediction abilities, their failure to account for epidemiological transmission mechanisms complicates the accurate interpretation of disease spread patterns. To tackle this issue, several researchers have incorporated compartmental models into Physics-Informed Neural Networks (PINNs). Their work indicates that this approach enables the estimation of temporal changes in model parameters with reasonable accuracy and provides dependable predictive methods [23–25]. This, in turn, offers plausible explanations for the efficacy of certain control strategies and the fundamental dynamics of epidemic transmission. For example, Kharazmi et al. applied PINNs to analyze various variants using the classic SIR model, with the aim of determining time-dependent parameters [26]. Ning et al. combined compartmental models with DNN to create Epidemiological priors informed deep neural networks (Epi-DNN), which aids in simulating the complex dynamics of COVID-19. They tested the effectiveness of Epi-DNN using real COVID-19 data from the Shanghai Omicron outbreak [27]. He et al. introduced Transmission Dynamics Informed Neural Networks (TDINN), which encode two versions of the compartmental model into DNN to predict time-dependent parameters [28,29]. They validated these predictions using real COVID-19 data from various regions and provided thoughtful interpretations of epidemic-related interventions. Saikia et al. utilized PINNs to analyze the ratio of undetected to detected active infections during the first and second waves of COVID-19 in India, giving a realistic picture of the Indian pandemic [30]. These studies demonstrate that the use of PINNs can estimate time-varying parameters within compartmental models with accuracy and offer a fresh perspective on understanding epidemic spread mechanisms, providing useful insights for the development of effective prevention and control strategies.

In this study, we introduced a new VOCs-INN algorithm, which integrates epidemiological data, deep learning, and epidemiological models to estimate the transmission intensity of various COVID-19 strains during the pandemic. It should be noted that the VOCs-INN algorithm emphasizes learning underlying patterns in multivariate data to improve model optimization and achieve accurate fitting. This algorithm is capable of fitting multi-source epidemic data effectively, and it can also be applied in scenarios involving a single virus. In such cases, the multi-strain model can be simplified to the classical SIR model with time-varying parameters, as proposed in 1927. Additionally, we applied this algorithm to analyze data from two VOIs in British Columbia (BC) and five internal health institutions in

Canada. The VOCs-INN algorithm also demonstrated good performance in simulating both observed and unobserved dynamics, facilitating time-dependent parameter inference and enabling short-term predictions. Furthermore, we compared the VOCs-INN algorithm with traditional fitting methods, showcasing its advantages in fitting accuracy and predictive performance. This analysis further supports the effectiveness and applicability of our proposed algorithm.

The remainder of this study is structured as outlined below. In the subsequent section, we will introduce the multi-strain infectious disease model and deep neural networks, establish the VOCs-INN algorithm, present the datasets utilized, and outline the experimental settings. Subsequently, we will share the key findings related to data fitting, parameter estimation, and predictions. We will also delve into the impact of time-varying transmission rates and the effective reproduction number on COVID-19 transmission, and validate the accuracy of the estimated parameters. Additionally, we will discuss comparisons between our proposed algorithm and traditional methods. Finally, in the discussion section, we will summarize our research findings.

## Methods

### Model formulation

Based on the fundamental *SIR* epidemiological model, we propose an $SI_1 \ldots I_n R$ epidemic model with VOCs. The transfer diagram is shown in Fig 1. Denote the number of VOCs by $n$, and the population is divided into susceptible individuals $S(t)$, infectious individuals $I_i(t)$ infected by the i-th VOC ($1 \leq i \leq n$), and recovered individuals $R(t)$. Additionally, we include auxiliary compartments, $I_i^c$, to keep track of the cumulative confirmed cases of infection by the i-th VOC. The parameters may change over time due to various factors, including government interventions, shifts in human behavior, virus mutations, and vaccination efforts. Therefore, to simulate the actual transmission dynamics of the pandemic accurately and effectively, we assume that the transmission rates and removal rates are functions of time, denoted by $\beta_i(t)$ and $\gamma_i(t)$. Table 1 describes the meaning of state variables and parameters.

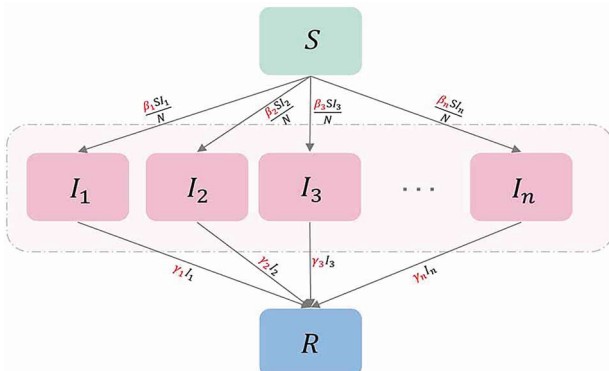

**Fig 1. The transfer diagram of model** (1).

**Table 1. Description of variables and parameters of model** (1).

| Symbol | Description | Unit | Range |
|---|---|---|---|
| **Variables** | | | |
| $t$ | Time | Week or day | \ |
| $S(t)$ | The number of susceptible individuals at time $t$ | Person | \ |
| $I_i(t)$ | The number of infected individuals with the i-th VOC at time $t$ | Person | \ |
| $R(t)$ | The number of removed individuals at time $t$ | Person | \ |
| $I_i^c(t)$ | The number of cumulative confirmed individuals with the i-th VOC at time $t$ | Person | \ |
| **Parameters** | | | |
| $N$ | Total population | Person | Fixed |
| $\beta_i(t)$ | The transmission rate of the i-th VOC at time $t$ | Per week or per day | $(0,1)$ |
| $\gamma_i(t)$ | The removal rate of the i-th VOC at time $t$ | Per week or per day | $(0,1)$ |

Consequently, the corresponding model is expressed as follows:

$$
\begin{cases}
\dfrac{\mathrm{d}S}{\mathrm{d}t} = -\sum_{i=1}^{n} \dfrac{\beta_i(t)SI_i}{N}, \\[2mm]
\dfrac{\mathrm{d}I_i}{\mathrm{d}t} = \dfrac{\beta_i(t)SI_i}{N} - \gamma_i(t)I_i, \quad i = 1, 2, \ldots, n, \\[2mm]
\dfrac{\mathrm{d}R}{\mathrm{d}t} = \sum_{i=1}^{n} \gamma_i(t)I_i, \\[2mm]
\dfrac{\mathrm{d}I_i^c}{\mathrm{d}t} = \dfrac{\beta_i(t)SI_i}{N}, \quad i = 1, 2, \ldots, n,
\end{cases}
\tag{1}
$$

with intital conditions $S(0) = S_0 > 0, I_i(0) = I_0^i > 0$, and $R(0) = R_0 \geq 0$. Here $N$ denotes the total population and

$$
N = S_0 + \sum_{i=1}^{n} I_0^i + R_0 = S(t) + \sum_{i=1}^{n} I_i(t) + R(t).
$$

Model (1) is both epidemiologically and mathematically well-posed.

We define the effective reproduction number for model (1) to be

$$
\mathcal{R}_e(t) = \max\{\mathcal{R}_{e,1}(t), \mathcal{R}_{e,2}(t), \ldots, \mathcal{R}_{e,n}(t)\},
$$

where

$$
\mathcal{R}_{e,i}(t) = \frac{\beta_i(t)S(t)}{\gamma_i(t)N}, \quad i = 1, 2, \ldots, n.
$$

Additionally, define the basic reproduction number by $\mathcal{R}_0 = \mathcal{R}_e(0)$. The basic reproduction number $\mathcal{R}_0$ represents the average number of secondary infections caused by a single infected individual in a susceptible population without any interventions. On the other hand, the effective reproduction number $\mathcal{R}_e$ dynamically quantifies the secondary infections caused by an infected individual in a population made up of both susceptible and non-susceptible host, and $\mathcal{R}_e$ decreases as $S(t)$ decreases. Therefore, both $\mathcal{R}_0$ and $\mathcal{R}_e$ are crucial indicators for policymakers in formulating prevention and control strategies. $\mathcal{R}_0$ helps in predicting the trajectory of an outbreak, while $\mathcal{R}_e$ provides real-time assessments of the effectiveness of

implemented measures. A high value of $\mathcal{R}_e$ indicates the need for intensified interventions, whereas sustained low levels may justify a gradual relaxation of restrictions.

### Physical information neural networks (PINNs)

Physics-informed neural networks [31] (PINNs) are data-driven neural network algorithms initially used to solve forward problems (approximate the solutions) and inverse problems (identify time-varying parameters) of partial differential equations (PDEs) [32–34]. In handling PDEs and time-varying parameters, PINNs have demonstrated unique advantages, making them highly promising for parameter inferences in epidemiological models. Recent studies [23–30] have shown that the PINNs framework can not only accurately fit data with compartmental models but also stably and effectively estimate time-varying parameters. Inspired by these studies, we propose a VOCs-INN method that can accurately fit data and infer complex time-varying parameters by encoding real VOCs infection data, epidemiological transmission mechanisms, and ODEs into neural networks.

Specifically, we designate $U_D^{NN}(t, \Theta_D)$ and $U_P^{NN}(t, \Theta_P)$ as two independent deep neural networks with the time variable $t$ as the input, and their weights and biases are parameterized by $\Theta_D$ and $\Theta_P$, respectively. Let

$$U_D^{NN}(t, \Theta_D) = \left( S^{NN}(t), I_1^{NN}(t), \ldots, I_n^{NN}(t), R^{NN}(t), I_1^{c\,NN}(t), \ldots, I_n^{c\,NN}(t) \right),$$

$$U_P^{NN}(t, \Theta_P) = \left( \beta_1^{NN}(t, \Theta_{\beta_1}), \ldots, \beta_n^{NN}(t, \Theta_{\beta_n}), \gamma_1^{NN}(t, \Theta_{\gamma_1}), \ldots, \gamma_n^{NN}(t, \Theta_{\gamma_n}) \right),$$

where $U_D^{NN}(t, \Theta_D)$ is used to fit data with the $SI_1 \ldots I_n R$ model (1) and approximate the solutions of the model; $U_P^{NN}(t, \Theta_P)$ comprises $2n$ independent deep neural networks, each utilized to identify different transmission rates $\beta_i(t)$ or removal rates $\gamma_i(t)$.

Fig 2 presents the visualization of the two aforementioned networks. Additionally, Fig 2 illustrates the loss composition of VOCs-INN. The loss of VOCs-INN has two parts. The first part, $LOSS_{data}$, is the mean squared error between the compartmental data outputted by network $U_D^{NN}(t, \Theta_D)$ and the real data:

$$
\begin{aligned}
Loss_{data} = {} & \frac{1}{N_d} \sum_{i=1}^{N_d} \left| S^{NN}(t_i) - S(t_i) \right|^2 + \frac{1}{N_d} \sum_{i=1}^{N_d} \left| R^{NN}(t_i) - R(t_i) \right|^2 \\
& + \sum_{j=1}^{n} \frac{1}{N_d} \sum_{i=1}^{N_d} \left| I_j^{NN}(t_i) - I_j(t_i) \right|^2 + \sum_{j=1}^{n} \frac{1}{N_d} \sum_{i=1}^{N_d} \left| I_j^{c\,NN}(t_i) - I_j^c(t_i) \right|^2 \\
& + \sum_{j=1}^{n} \frac{1}{N_d} \sum_{i=1}^{N_d} \left| I_j^{n\,NN}(t_i) - I_j^n(t_i) \right|^2,
\end{aligned}
\tag{2}
$$

where $N_d$ represents the number of training data, and

$$I_j^{n\,NN}(t) = I_j^{c\,NN}(t) - I_j^{c\,NN}(t-1)$$

represents the newly confirmed infected individuals. It should be noted that while $Loss_{data}$ includes calculations for data from different compartments, only some compartments have

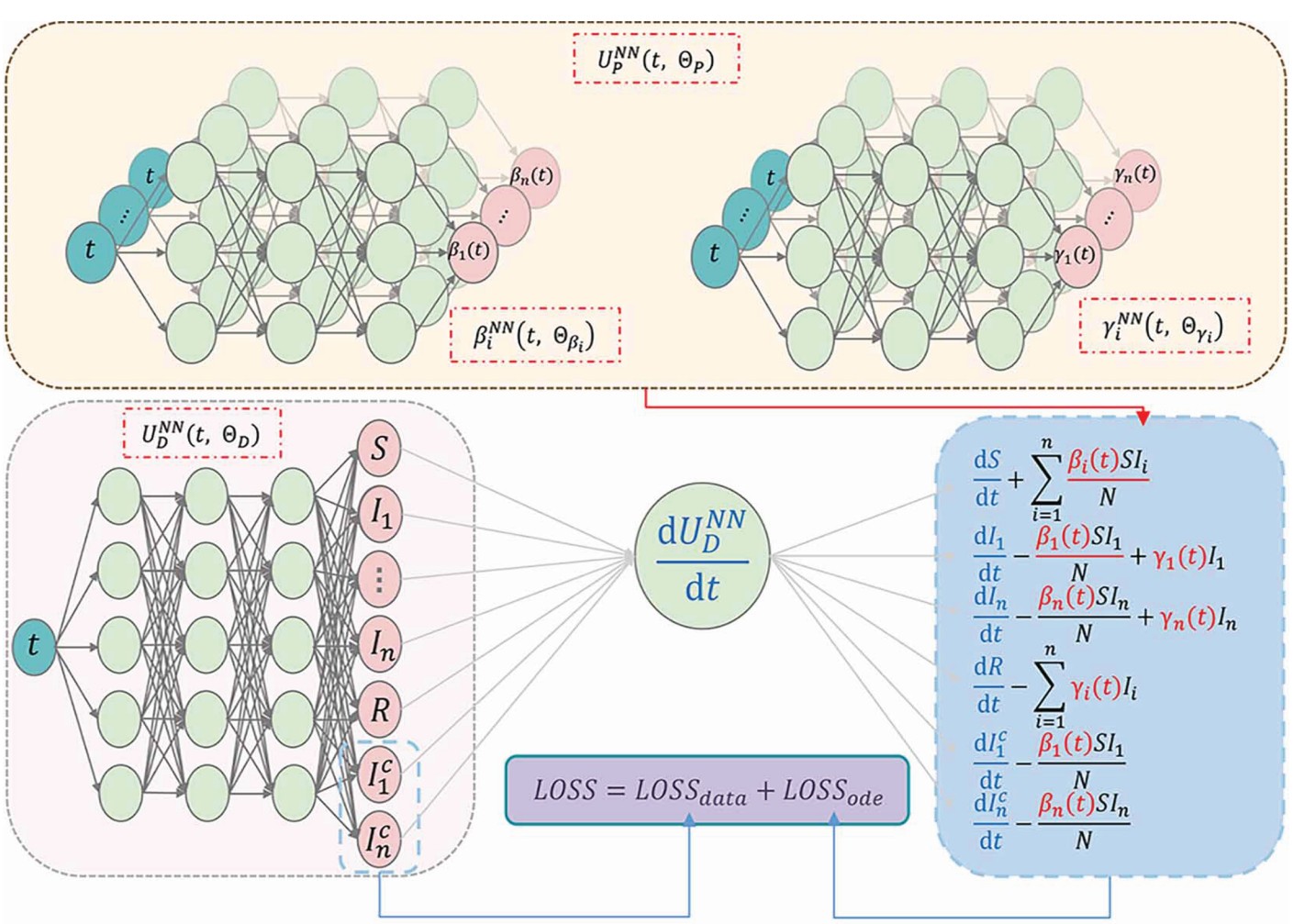

**Fig 2. Schematic diagram of VOCs-INN for $SI_1 \ldots I_nR$ model.** $U_D^{NN}(t, \Theta_D)$ is used to fit the state variables of the model (1) (represented by the pink shaded area), while $U_P^{NN}(t, \Theta_P)$ is employed to infer the time-varying parameters (represented by the yellow shaded area). Specifically, $\beta_i^{NN}(t, \Theta_{\beta_i})$ and $\gamma_i^{NN}(t, \Theta_{\gamma_i})$ are used to infer $\beta_i(t)$ and $\gamma_i(t)$, respectively. $\frac{dU_D^{NN}}{dt}$ denotes the automatic differentiation operator, and the $LOSS$ consists of two components, $LOSS_{data}$ and $LOSS_{ode}$. By minimizing the $LOSS$, simultaneous data fitting and inference of time-varying parameters can be achieved.

real data support. Therefore, it is necessary to adjust the calculation of $Loss_{data}$ based on available data.

Additionally, the solutions outputted by $U_D^{NN}(t, \Theta_D)$ and the time–varying parameters identified by network $U_P^{NN}(t, \Theta_P)$ should adhere to the ordinary differential equations defined by $SI_1 \ldots I_nR$ model. Therefore, the second part of the loss is defined as $LOSS_{ode}$, serving as a penalty term to ensure that the neural network outputs conform to the ordinary differential equations. The form of $LOSS_{ode}$ is as follows:

$$Loss_{ode} = \frac{1}{N_e} \sum_{k=1}^{4} \sum_{i=1}^{N_e} |L_k(t_i)|^2, \tag{3}$$

where $N_e$ represents the number of residual points, which are randomly selected from the entire computational domain and

$$L_1\left(t_i\right) = \frac{\mathrm{d}S^{NN}\left(t_i\right)}{\mathrm{d}t} + \frac{1}{N}\sum_{j=1}^{n}\beta_j^{NN}\left(t_i\right)S^{NN}\left(t_i\right)I_j^{NN}\left(t_i\right),$$

$$L_2\left(t_i\right) = \sum_{j=1}^{n}\left[\frac{\mathrm{d}I_j^{NN}\left(t_i\right)}{\mathrm{d}t} - \left(\frac{1}{N}\beta_j^{NN}\left(t_i\right)S^{NN}\left(t_i\right) + \gamma_j^{NN}\left(t_i\right)\right)I_j^{NN}\left(t_i\right)\right],$$

$$L_3\left(t_i\right) = \frac{\mathrm{d}R^{NN}\left(t_i\right)}{\mathrm{d}t} - \sum_{j=1}^{n}\gamma_j^{NN}(t_i)I_j^{NN}(t_i),$$

$$L_4\left(t_i\right) = \sum_{j=1}^{n}\left[\frac{\mathrm{d}I_j^{cNN}\left(t_i\right)}{\mathrm{d}t} - \frac{1}{N}\beta_j^{NN}\left(t_i\right)S^{NN}\left(t_i\right)I_j^{NN}(t_i)\right].$$

(4)

By continuously minimizing the loss function

$$LOSS = LOSS_{data} + LOSS_{ode}$$

to train the deep neural network, we have determined the optimal fitting network parameters $\Theta_D^*$ and $\Theta_P^*$. With these optimal network parameters $\Theta_D^*$ and $\Theta_P^*$, we have effectively achieved data fitting and identification of time-varying parameters.

## Experiment settings

We collected COVID-19 datasets from the province of British Columbia (BC) Centre for Disease Control (CDC) in Canada [35]. BC and its five regions datasets from March 26, 2023, to January 7, 2024, are shown in Figs 3 and Supporting information, respectively. During this period, two main VOIs were XBB.1.16.* and XBB.1.5.*. They are both mutant variants of the XBB family and have inherited some of its characteristics [36]. The XBB. 1.5 and XBB. 1.5.* variants were first discovered in Australia, India, and Bangladesh, while the XBB. 1.16 and XBB. 1.16.* variants were first discovered in India [37]. They emerged independently from one another, and each rapidly became dominant, regionally or globally, outcompeting previous variants. They also show distinct levels of virulence, antigenicity, and infectivity. Our datasets are limited to the weekly new infection data $(I_1^n, I_2^n)$ and weekly cumulative infection data $(I_1^c, I_2^c)$ of XBB.1.16.* and XBB.1.5.* respectively.

To control the spread of the COVID-19 pandemic, the BC government implemented a series of measures. These measures were accompanied by the release of related reports, which can be organized into a timeline depicted in Fig 4. In Fig 4, the five nodes marked with large circles represent extremely important events or milestones. These nodes are the core of the analysis and have a significant impact on overall trends or results. In contrast, the other five nodes marked with small circles, while still important, have relatively low importance and may represent secondary events or transitional stages on the timeline. All reports and intervention measures are publicly accessible through official websites, providing a comprehensive record for analytical and research purposes.

When fitting the data with the $SI_1I_2R$ model, there are four time-dependent parameters $\beta_1(t), \beta_2(t)$ and $\gamma_1(t), \gamma_2(t)$ need to infer. The initial values of each compartment and the prior recovery rate ranges are summarized in Table 2. It is worth noting that the initial value $R(0)$ for the removal population is assumed to be 0, which is mainly based on the XBB variant's high immune escape ability. In addition, the data period we fitted here is nearly nine

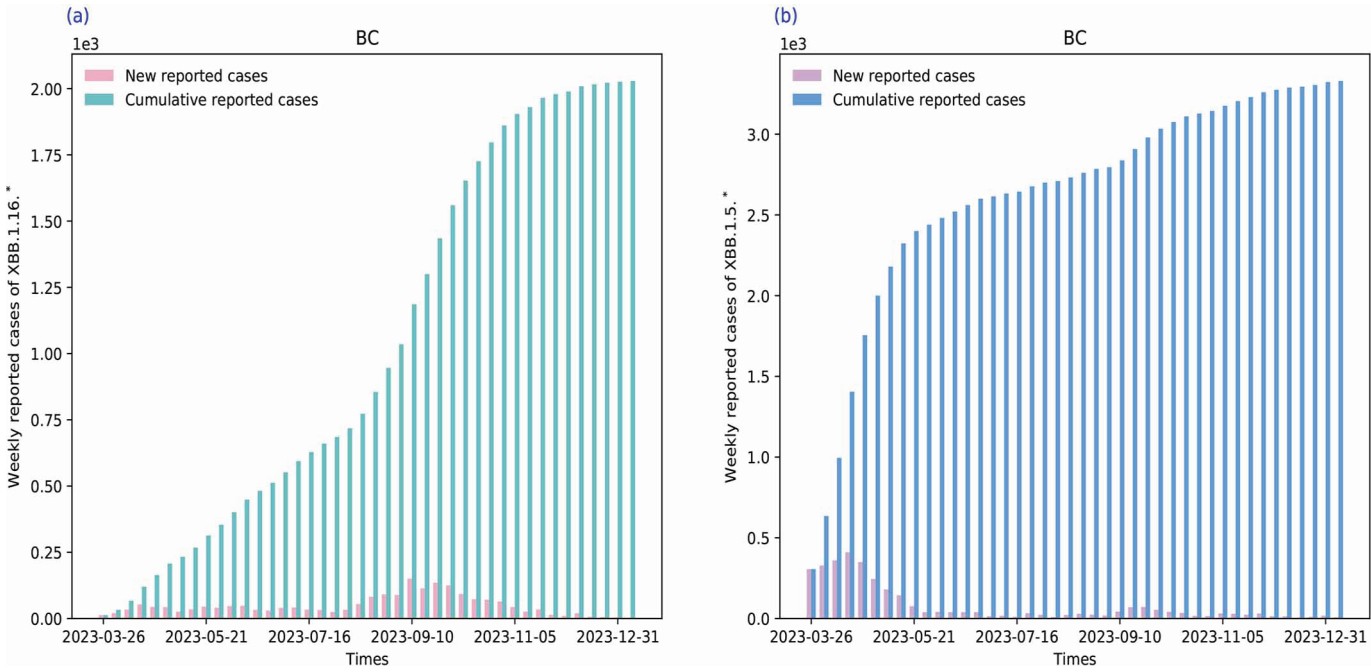

**Fig 3**. **Datasets for** $SI_1I_2R$ **model.** The graph on the left shows weekly new confirmed and weekly cumulative infected data for XBB.1.16.* from 26 March 2023 to 7 January 2024. The graph on the right shows weekly new confirmed and weekly cumulative data for XBB.1.5.* from 26 March 2023 to 7 January 2024. Both graphs have the horizontal axis in days and the vertical axis in number of people.

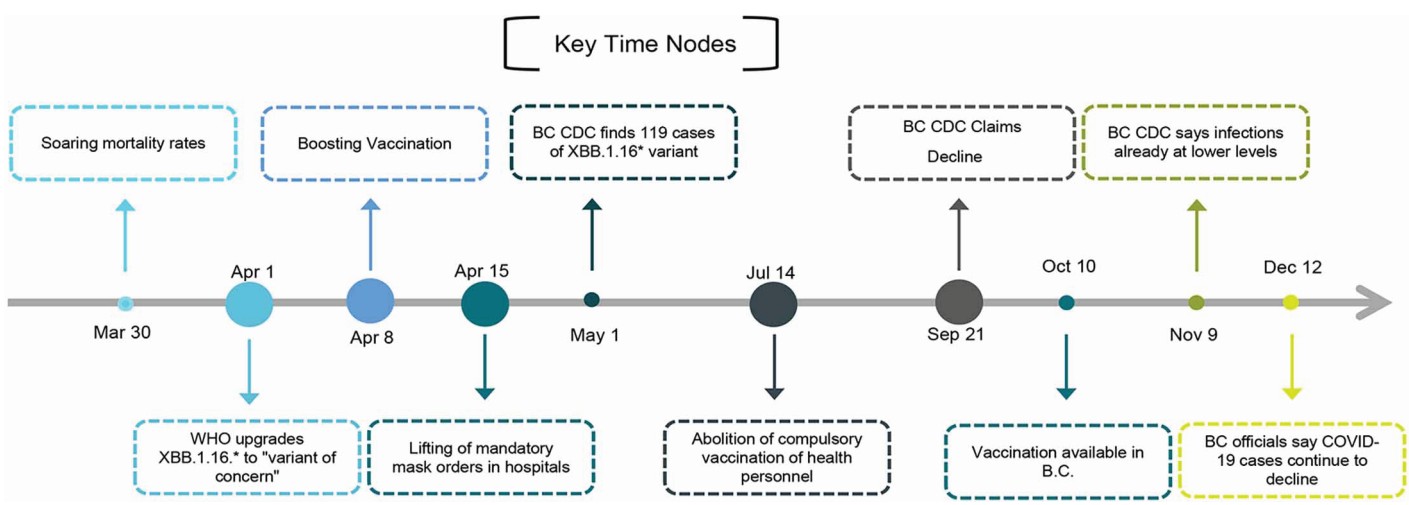

**Fig 4**. **Timeline of non-pharmaceutical interventions (NPIs) implemented in BC to control COVID-19** [38–42].

months, far beyond the safe window period after infection. We predefine hyperparameters to effectively train the neural network and summarize them in Table 3. As shown in Table 3, we present the hyperparameter values of the deep neural networks $U_D^{NN}(t)$, $\beta_1^{NN}(t)$, $\beta_2^{NN}(t)$, $\gamma_1^{NN}(t)$ and $\gamma_2^{NN}(t)$ for different regions. For instance, $U_D^{NN}(t) : 5 * 32$ for the BC province indicates that the neural network has five hidden layers, each containing 32 neurons. During training, the Adam and L-BFGS optimizers optimize the neural network parameters to

**Table 2. The known ranges of the parameters and initial values of compartments.**

| Regions | BC | Interior (IH) | Fraser (FH) | Northern (NH) | Vancouver Coastal (VCH) | Island (VIH) |
|---|---|---|---|---|---|---|
| $\gamma_1, \gamma_2$ | (0.49, 0.51) | 0.5 | 0.5 | 0.5 | 0.5 | 0.5 |
| $N$ | $5.5 \times 10^6$ | $1.03 \times 10^6$ | $2.05 \times 10^6$ | $1.25 \times 10^6$ | $9 \times 10^5$ | $2.7 \times 10^5$ |
| $S(0)$ | $5.5 \times 10^6$ | $1.03 \times 10^6$ | $2.05 \times 10^6$ | $1.25 \times 10^6$ | $9 \times 10^5$ | $2.7 \times 10^5$ |
| $I_1(0)$ | 13 | 2 | 6 | 1 | 3 | 1 |
| $I_2(0)$ | 306 | 43 | 79 | 80 | 86 | 13 |
| $R(0)$ | 0 | 0 | 0 | 0 | 0 | 0 |

**Table 3. The values of hyperparameters in deep neural networks in different regions.**

| Regions | BC | Interior (IH) | Fraser (FH) | Northern (NH) | Vancouver Coastal (VCH) | Island (VIH) |
|---|---|---|---|---|---|---|
| Scale of $U_D^{NN}(t)$ | 5*32 | 3*32 | 5*32 | 3*32 | 3*16 | 3*32 |
| Scale of $\beta_1^{NN}(t)$ | 3*64 | 5*32 | 3*64 | 5*32 | 3*32 | 5*32 |
| Scale of $\gamma_1^{NN}(t)$ | 3*64 | 5*32 | 3*64 | 5*32 | 3*32 | 5*32 |
| Scale of $\beta_2^{NN}(t)$ | 3*64 | 5*32 | 3*64 | 5*32 | 3*32 | 5*32 |
| Scale of $\gamma_2^{NN}(t)$ | 3*64 | 5*32 | 3*64 | 5*32 | 3*32 | 5*32 |
| Iterations | $1 \times 10^4$ | $1 \times 10^4$ | $1 \times 10^4$ | $1 \times 10^4$ | $1 \times 10^4$ | $1 \times 10^4$ |

minimize the loss function *LOSS*. Firstly, the neural network is optimized using the Adam optimizer, and the number of iterations for the Adam optimizer is provided in Table 3, with a fixed learning rate of $1 \times 10^{-4}$. Subsequently, the L-BFGS optimizer is used to further optimize the model, with a maximum of $5 \times 10^4$ iterations, terminating early if the default conditions are met. Based on our datasets, a more precise VOCs-INN framework is shown in Supporting information. The hyperbolic tangent function Tanh [44] is also used as the activation function. The VOCs-INN algorithm utilizes the open-source library TensorFlow-GPU 1.14 [45] in Python for automatic differentiation and deep learning computations.

## Results

By combining VOCs-INN with infectious disease models, we have been able to fit epidemic data accurately and estimate parameters effectively. This method shows the time-varying nature of the parameters and can also make short-term predictions, making it useful for real-time disease monitoring and control. VOCs-INN also demonstrates its strong ability to learn and capture the underlying patterns of the dynamics.

### Data fitting and parameter inferences

We used the $SI_1I_2R$ model and datasets from BC and its five regions to train our VOCs-INN. The fitting results for the weekly and cumulative reported cases for the two VOIs of BC are presented in Fig 5. Additionally, fitting results for the five inner regions are provided in Supporting information, and extrapolation results for the unobserved data are available in Supporting information. In Fig 5, we show the time-dependent parameter estimation results. The red dashed lines represent the fitted curves for the transmission rates $\beta_1(t)$ and $\beta_2(t)$, while the blue dashed lines represent the fitted curves for the removal rates $\gamma_1(t)$ and $\gamma_2(t)$.

To quantify the goodness-of-fit of our model, we use the coefficient of determination ($R^2$), which is a statistical measure of how observed outcomes are replicated by the model, based on the proportion of total variation of outcomes [46]. The $R^2$ is less than 1, and an $R^2$ of 1

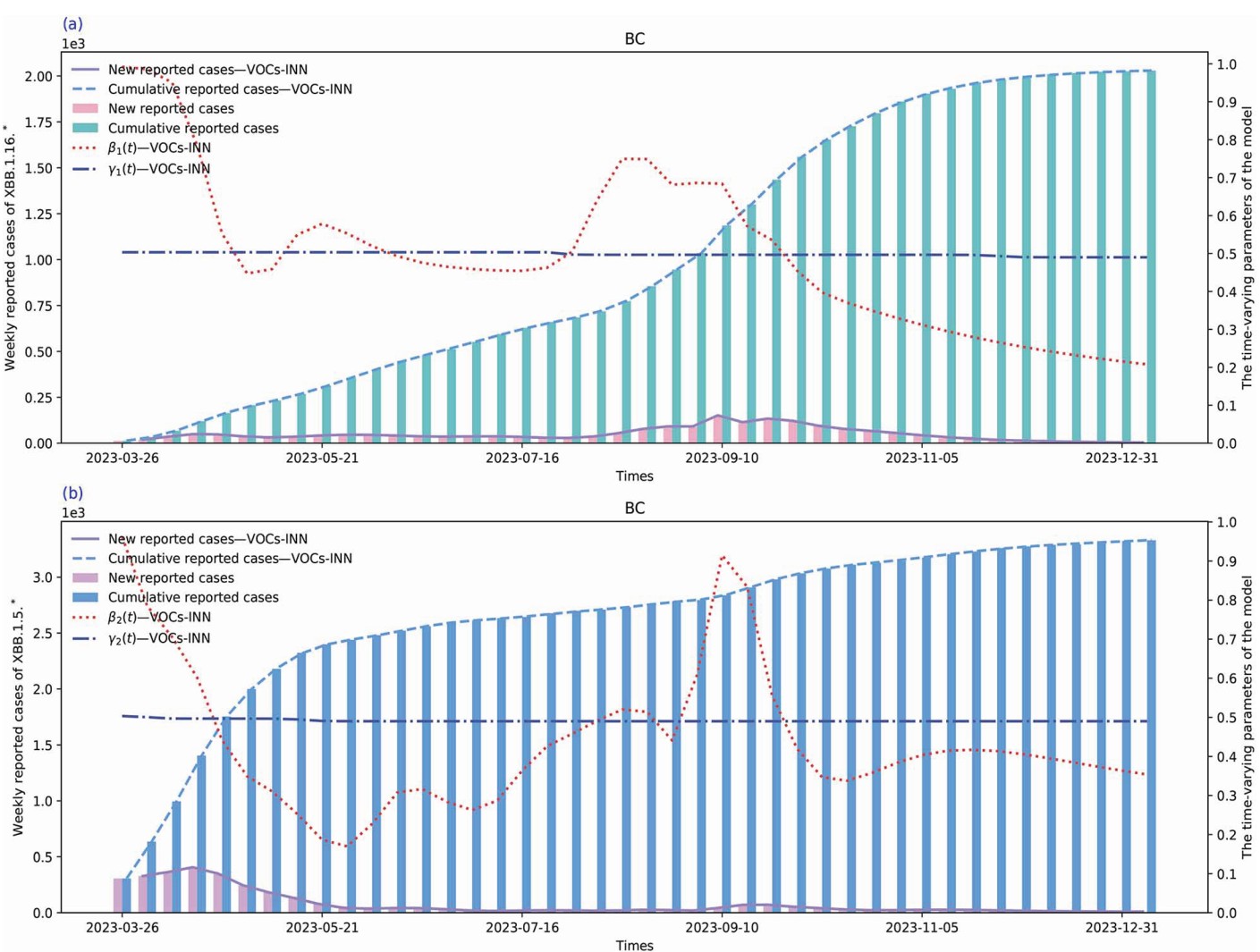

**Fig 5. Data fitting and inferences of time-dependent parameters for BC using the VOCs-INN algorithm.** (a) Two regional blocks represent newly confirmed cases and cumulative confirmed cases. The red and dark blue dashed lines represent the values of $\beta_1(t)$ and $\gamma_1(t)$ inferred using VOCs-INN, while the purple and light blue lines represent newly confirmed cases and cumulative confirmed cases fitted using VOCs-INN. (b) Similar to the above. Both graphs have the horizontal axis in days and the vertical axis in number of people.

indicates that the model predictions perfectly fit the data. The $R^2$ for our model can be expressed mathematically as follows:

$$R^2 = 1 - \frac{\sum_{i=1}^{n}(y_i - \hat{y}_i)^2}{\sum_{i=1}^{n}(y_i - \bar{y})^2}.$$

(5)

Here, $y_i$ represents the observed values, $\hat{y}_i$ represents the fitted values, $\bar{y}$ is the mean of the observed values, and $n$ is the number of observations. Based on our fitting results in Fig 5, we calculate the $R^2$ values of two fitting sets and obtain $R^2$ of 0.9979 for XBB. 1.16.* and 0.9865 for XBB. 1.5.*. These indicate that our model predictions fit the data almost perfectly. Also, they confirm the reasonableness and creditability of our parameter estimations. When comparing the estimated parameters with the actual data, we find considerable synchronism

between the transmission rates and the fluctuations in the data. It was observed that there is always a time lag between the peak of transmission rate and the peak of reported cases per day in [29]. However, our results do not demonstrate this phenomenon, primarily because we use weekly data. Furthermore, the removal rates $\gamma_i(t)$ are relatively stable; this is because $\frac{1}{\gamma_i(t)}$ are the average infectious periods of the two VOIs. We also observe that the curve of $\beta_1$ is flatter than that of $\beta_2$, while the fluctuations of $\beta_2$ are more noticeable, and the value of $\beta_2$ at the end of time is more significant than that of $\beta_1$. This suggests that XBB.1.5.* was more dominant during this period, which is consistent with the findings of [8].

Based on the estimated time-dependent functions of the parameters, we have calculated the basic reproduction numbers, $\mathcal{R}_{0,1}$ and $\mathcal{R}_{0,2}$, as well as the effective reproduction numbers, $\mathcal{R}_{e,1}$ and $\mathcal{R}_{e,2}$, for two VOIs of BC. These results are presented in Fig 6. The corresponding findings for the other five internal regions are shown in Supporting information. The estimated values for the basic reproduction numbers are approximately $\mathcal{R}_{01} = 1.9710$ and $\mathcal{R}_{02} = 1.9108$. Additionally, the effective reproduction numbers vary over time with $\mathcal{R}_{e1} \in [0.4235, 1.9710]$ and $\mathcal{R}_{e2} \in [0.3463, 1.9108]$.

### Inference comparison of BC province with its five internal regions

The estimated time-varying transmission rates of two VOIs in BC and its five internal regions using VOCs-INN are depicted in Fig 7. We have the following observations:

- Three internal regions of the BC province, FH, VCH, and VIH, display a similar overall trend to the BC province. [47] provides a visualization of the geographical locations of BC province. This similarity is likely due to their close geographical proximity. FH, VCH, and VIH are located near the coast and distant from the inland, and their proximity to each other may result in higher similarities and consistency in terms

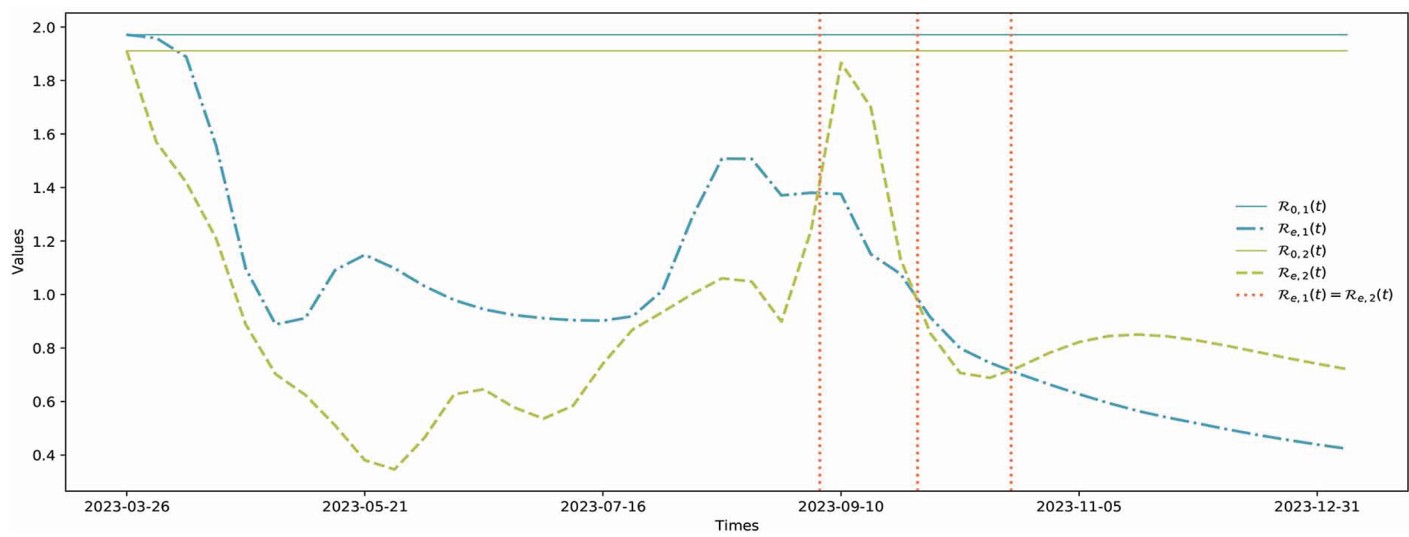

**Fig 6. Comparisons of the basic reproduction numbers and effective reproduction numbers of the two VOIs.** The two blue lines represent the basic reproduction numbers and effective reproduction numbers of XBB.1.16.* variant, the two yellow lines represent the basic reproduction numbers and effective reproduction numbers of XBB.1.5.* variant, and the red dashed line represents the value when the effective reproduction numbers of the two VOIs are the same.

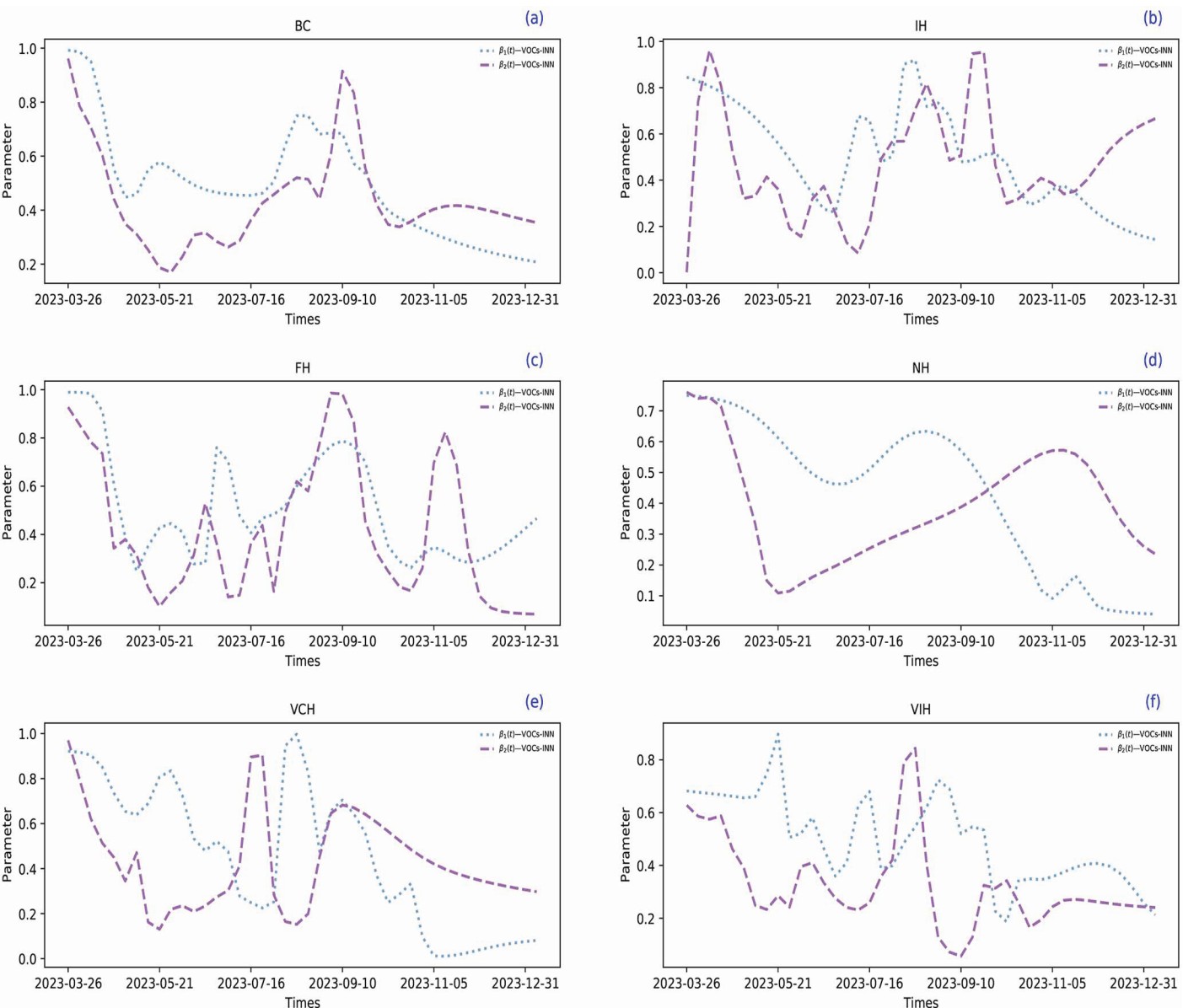

**Fig 7. The inference plots for $\beta_1(t)$ and $\beta_2(t)$ in BC province and its five internal regions.** The blue dashed line represents the transmission rate of the XBB.1.16.\*, while the purple dashed line depicts the transmission rate of the XBB.1.5.\*.

of personnel mobility, economic activities, and the implementation of epidemic prevention measures. Consequently, their transmission rates and epidemic progression patterns tend to be similar.

- The trend observed in the IH region differs notably from those in the BC province and other regions. IH is located inland and shares a border with the province of Alberta, which has implemented strict prevention and control policies. This geographical and policy divergence may contribute to the differences in epidemic transmission and

prevention measures between IH and other regions. Additionally, due to its proximity to Alberta, IH may be influenced by its prevention policies, resulting in a unique epidemic development trend.

- The trend of change in the NH region seems to be somewhat slower compared to other regions. This could be attributed to NH being the largest region in the BC province with a low population density. In areas with low population density, the speed and scope of epidemic transmission may be influenced by various factors, such as population density, personnel mobility, and medical resources. Therefore, NH may face more significant challenges in responding to the epidemic, leading to a relatively slower change in its transmission rate.

The transmission rates in the BC province are influenced by the interactions among its five internal regions. These findings indicate the complexity and variety of how infectious diseases are transmitted and spread.

## Prediction

In this subsection, we evaluate the performance of our proposed VOCs-INN algorithm in forecasting the future trend of the COVID-19 pandemic. We assign the transmission rates and removal rates for the subsequent 16 weeks (from 7 January to 26 April 2024) to the values inferred by the VOCs-INN algorithm at the final time point (7 January 2024), as illustrated in Supporting information (For the transmission rate prediction maps of the five regions, please refer to Supporting information). Using these parameters, we predict the weekly reported cases and cumulative number of infections in BC over the next 16 weeks, as shown in Figs 8 and 9, respectively. The forecasted results for the five regions within BC are presented in Supporting information. Given the lack of information on the data for the confidence intervals, we incorporate uncertainty bounds of 10% and 20% to account for the impact of various control measures on the time-dependent parameters $\beta_i(t)$ and $\gamma_i(t)$, and then propagate this uncertainty to the prediction component. By comparing the predictions with the unused data from the fitting process in Figs 8 and 9, we observe that the time-dependent $SI_1I_2R$ model calibrated by the VOCs-INN algorithm predicts the trend of future outbreaks quite well, with the shaded areas differing by a small number of people from the real data.

## Verification

To verify the effectiveness of training, fitting, and estimation results of our proposed VOCs-INN algorithm, we incorporate the time-dependent transmission rates $\beta_i(t)$ and $\gamma_i(t)$ estimated by the VOCs-INN algorithm, along with other constant parameters, into the $SI_1I_2R$ model. We then utilize the ODE solver in Python to solve the model. In Figs 10 and 11, using the numerical solutions obtained from our mathematical model, we present the weekly and cumulative reported cases for two VOIs. The simulation curves align well with the data. This indicates that the training results are consistent with the transmission mechanisms identified by the mathematical model (see Supporting information for images of the five regions).

## Comparison with nonlinear least squares method

To validate the effectiveness of our proposed VOCs-INN, we selected the traditional nonlinear least squares method as a benchmark for comparison. The experimental setup is as follows. We chose the classical SIR model as epidemiological information, i,e. model (1) with a single strain. In the data preparation phase, we selected daily new COVID-19 case data from

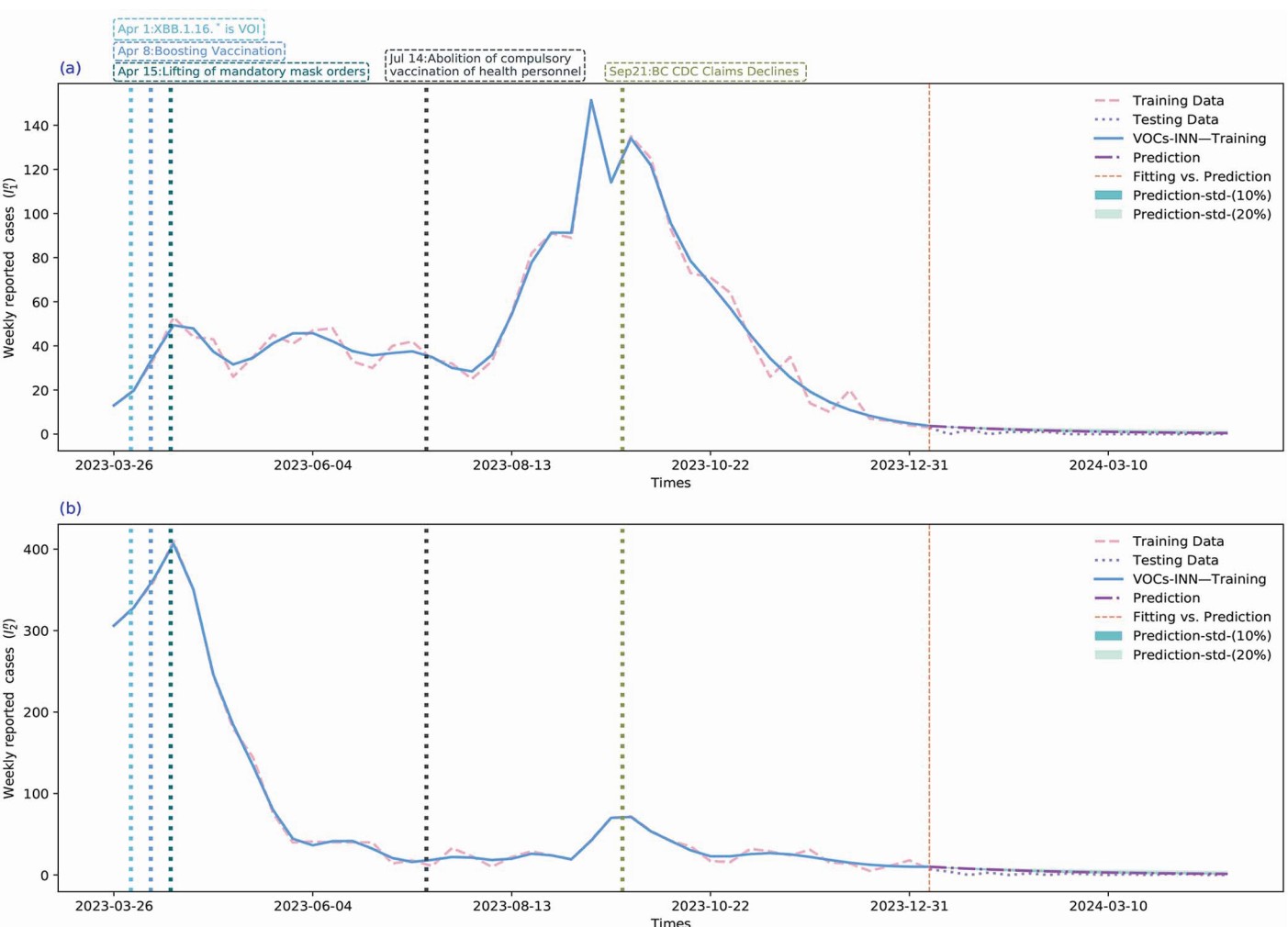

**Fig 8. Predictions of weekly newly reported cases of BC for the upcoming 16 weeks since 7 January 2024 using a two-VOCs model learned by the VOCs-INN algorithm.** The red dashed lines represent the training data, the lavender dotted lines represent the validation data, and the purple dashed lines represent the predictions. The shaded areas indicate the uncertainty range obtained by adjusting the transmission rates up or down by 10% and 20%. The enlarged image of the shaded area can be found in Supporting information. The orange vertical dashed line represents the segmentation line between fitting and prediction.

Italy between November 7, 2021, and June 28, 2022, as our fitting dataset, while data from June 28, 2022, to July 12, 2022, were used to assess prediction performance.

Initially, we attempted to estimate the transmission rate $\beta$ and removal rate $\gamma$ by fitting the *SIR* model to the dataset using the nonlinear least squares method, and the fitting results are shown in Fig 12(a). This figure shows that the real data has multiple peaks, but the least squares fitting result only captures one peak, suggesting that the fitting is not good. This issue is mainly because the transmission rate $\beta$ is treated as a constant in the least squares method, making it difficult to simulate multiple peaks accurately.

To enhance the fitting effect, we divided the fitting data into three subsets for piecewise nonlinear least squares estimation based on the observed peak patterns in the dataset. The results of piecewise estimation are presented in Fig 12(b). From this figure, we can see that while the piecewise estimation method improves the fitting of peak times and overall trends, some deviations remain. Next, we used the best-fit parameter values of the third stage to solve

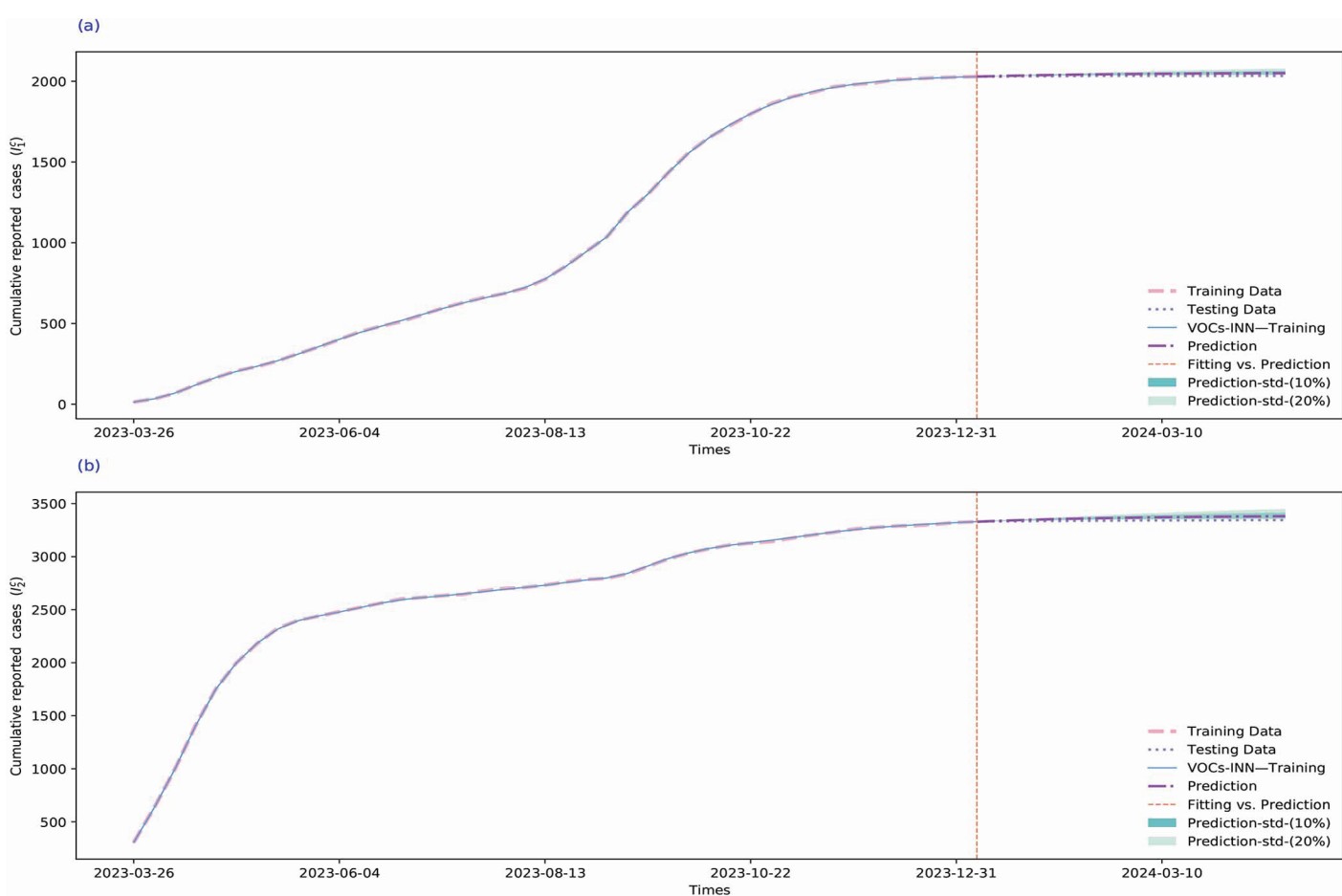

**Fig 9. Cumulative weekly reported cases of BC for the upcoming 16 weeks from 7 January 2024 (i.e., vertical line), as predicted using a two-VOCs model learned by the VOCs-INN algorithm.** The red dashed line represents the training data, the lavender dotted line represents the validation data, and the purple dashed lines represent the predictions. The shaded areas indicate the uncertainty range obtained by adjusting the transmission rates up or down by 10% and 20%. The enlarged image of the shaded area can be found in Supporting information. The orange vertical dashed line represents the segmentation line between training data and validation data.

the ODE model and predict the daily reported cases over the coming two weeks. The prediction results show that this method is able to capture the increasing trend of daily reported cases, but there is still room for improvement in prediction accuracy.

Although the piecewise estimation does improve the fitting effect to a certain degree, the dataset's segmentation relies on manually identified peak trends, which introduces a significant subjective component. To address this issue, we used the VOCs-INN model for fitting and prediction, shown in Fig 12(c). To compare the fitting and prediction performance of the three methods comprehensively, we calculated their Mean Squared Error (MSE) and coefficient of determination ($R^2$), with the specific results listed in Table 4. By examining the data in Table 4, it is evident that the VOCs-INN model performs better than the traditional methods in both fitting and prediction.

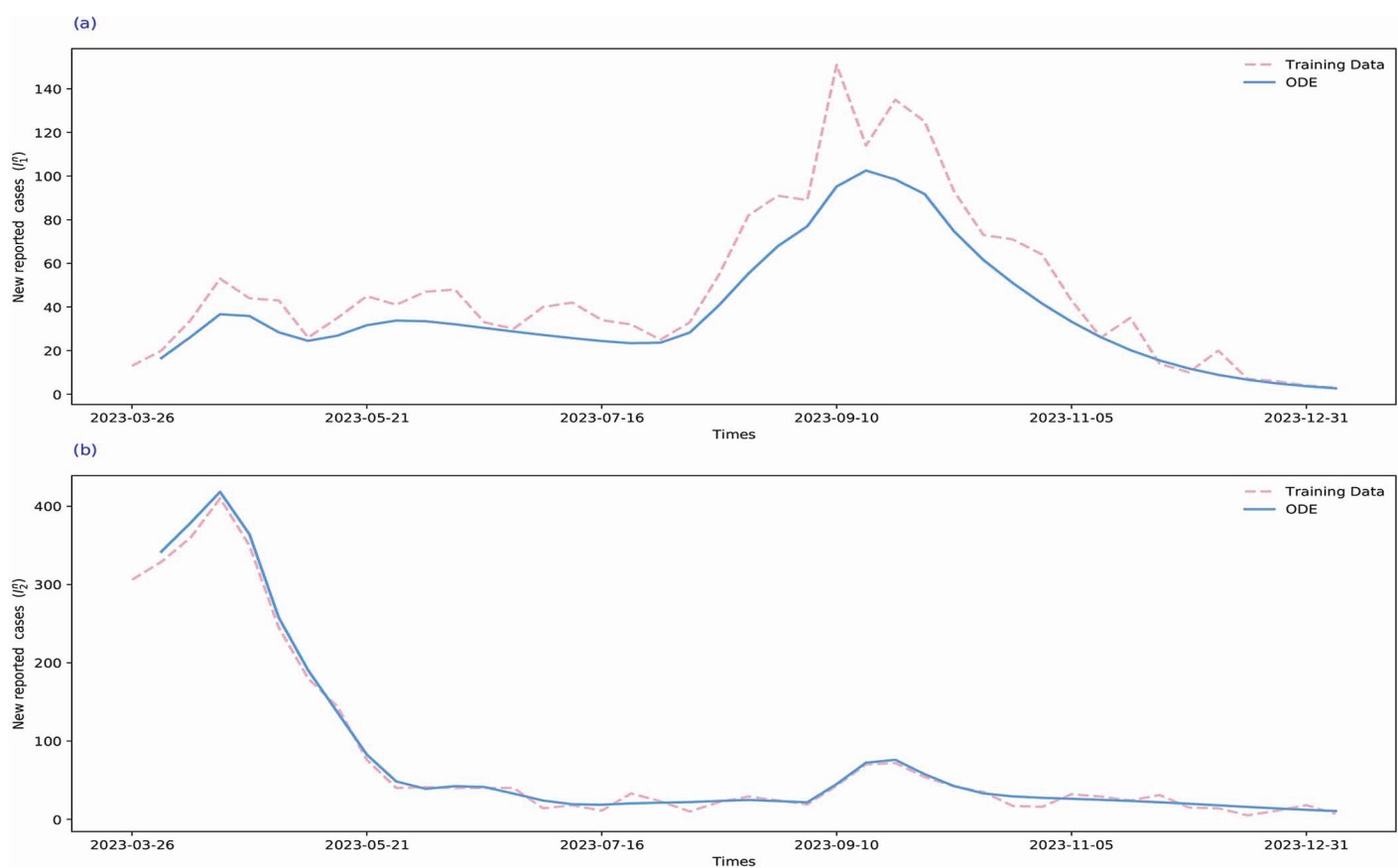

**Fig 10**. **The $SI_1I_2R$ model is numerically solved by replacing the parameters with the VOCs-INN inferences.** The pink dashed line and blue curve represent the data on new cases reported per week and the corresponding numerical solution for new cases reported per week.

## The impact of physical information

Furthermore, to gain a deeper understanding of the impact of physical information on model performance, we designed the following experiment. By gradually adjusting the weight ratio of $LOSS_{ode}$ relative to $LOSS_{data}$, and carefully observing the changes in the goodness of fitting as the weight ratio increased from 0 to 1. Specifically, we conducted experiments with five weight values: $0, 0.25, 0.5, 0.75$, and $1$. The results of these experiments are presented in Fig 13. A weight of 0 means that only neutral networks are used for fitting without any physical information. In this case, as illustrated in Fig 13(e), the data fitting seems perfect, however the solution to the ODE doesn't match the training data, especially during the peak period. As the weight gradually increased from a non-zero value to 1, we observed that the ODE inversion effect gradually improved, while the fitting effect remained relatively stable. This observation underscores the significance of weight selection. To improve the overall performance and prediction accuracy of the algorithm, it is necessary to find a suitable weight ratio between $LOSS_{ode}$ and $LOSS_{data}$. However, it is important to note that this does not necessarily imply that a higher weight is always advantageous. Excessively high weights may overly emphasize physical information constraints while overlooking the inherent characteristics of the data, potentially resulting in degraded performance in data fitting. Therefore, finding this balance

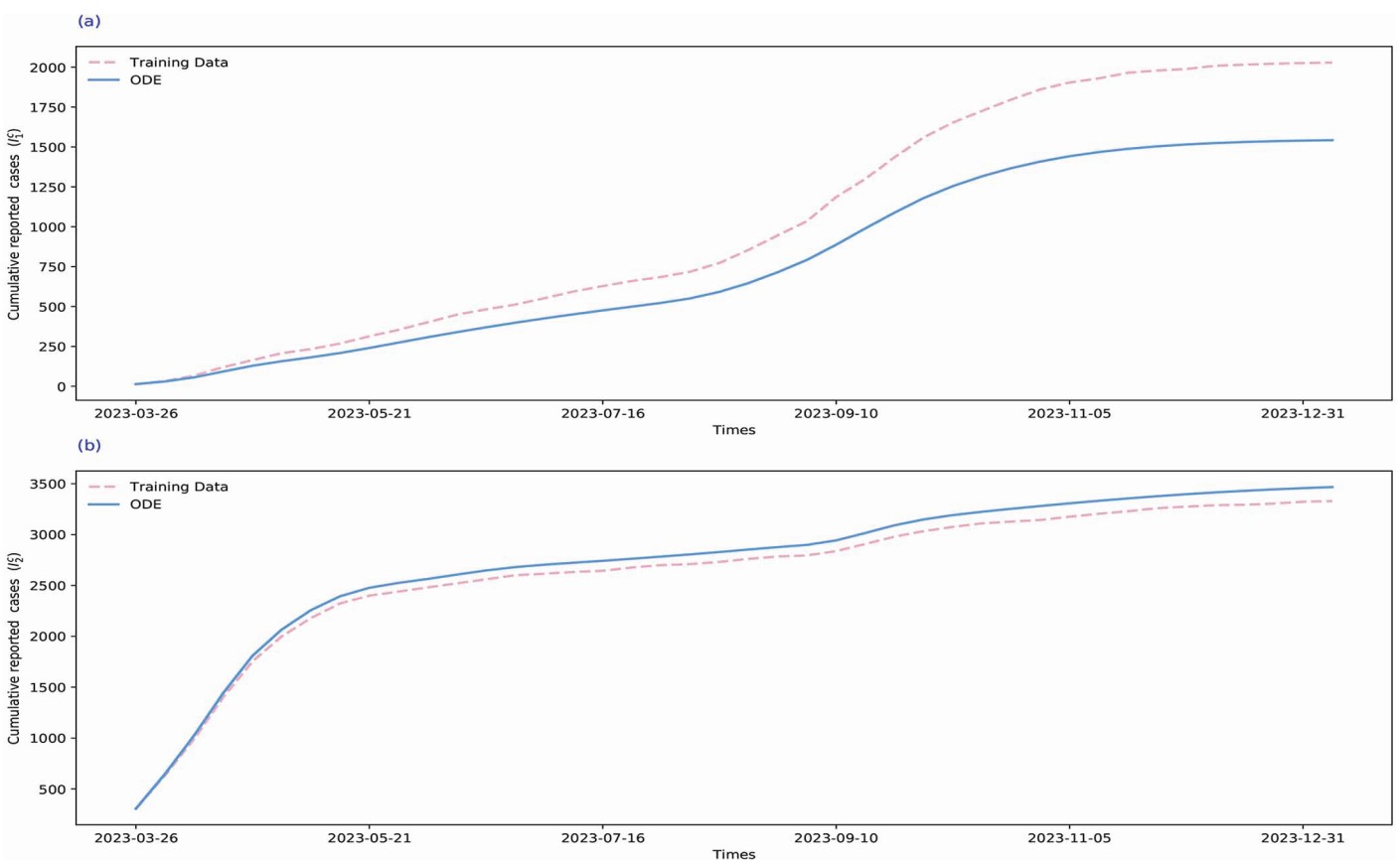

**Fig 11**. **The $SI_1I_2R$ model is numerically solved by replacing the parameters with the VOCs-INN inferences.** The pink dashed line and blue curve represent the data on cumulative reported cases per week and the corresponding numerical solution for cumulative reported cases per week.

requires a thorough consideration of various factors, including data characteristics, the physical background of the model, and the requirements of practical applications. The precise determination of this balance point is essential for ensuring the validity and reliability of the model.

## Discussion

In our research, we sought to address the complexities of understanding and modeling the transmission dynamics of Variants of Concern (VOCs) by integrating an $SI_1 \ldots I_nR$ model with neural networks. This innovative approach, termed VOCs-INN, allows the analysis of time-varying characteristics of VOCs, such as transmission rates, removal rates, and effective reproduction numbers.

The VOCs-INN algorithm was applied to study the spread of two VOCs in the province of BC and its five constituent regions. Our research results indicate that the fitting results of this algorithm align well with real-world data, enabling accurate short-term disease progression predictions. The parameters derived from the neural network are persuasive, and inverse simulations using an ODE solver further validate the accuracy of our estimations.

The alignment of our estimated effective reproduction number in BC province with figures reported in related research articles provides reassurance about the validity of our approach. For instance, Manathunga, S. S et al. [48] used a local regression (LOESS) model to estimate

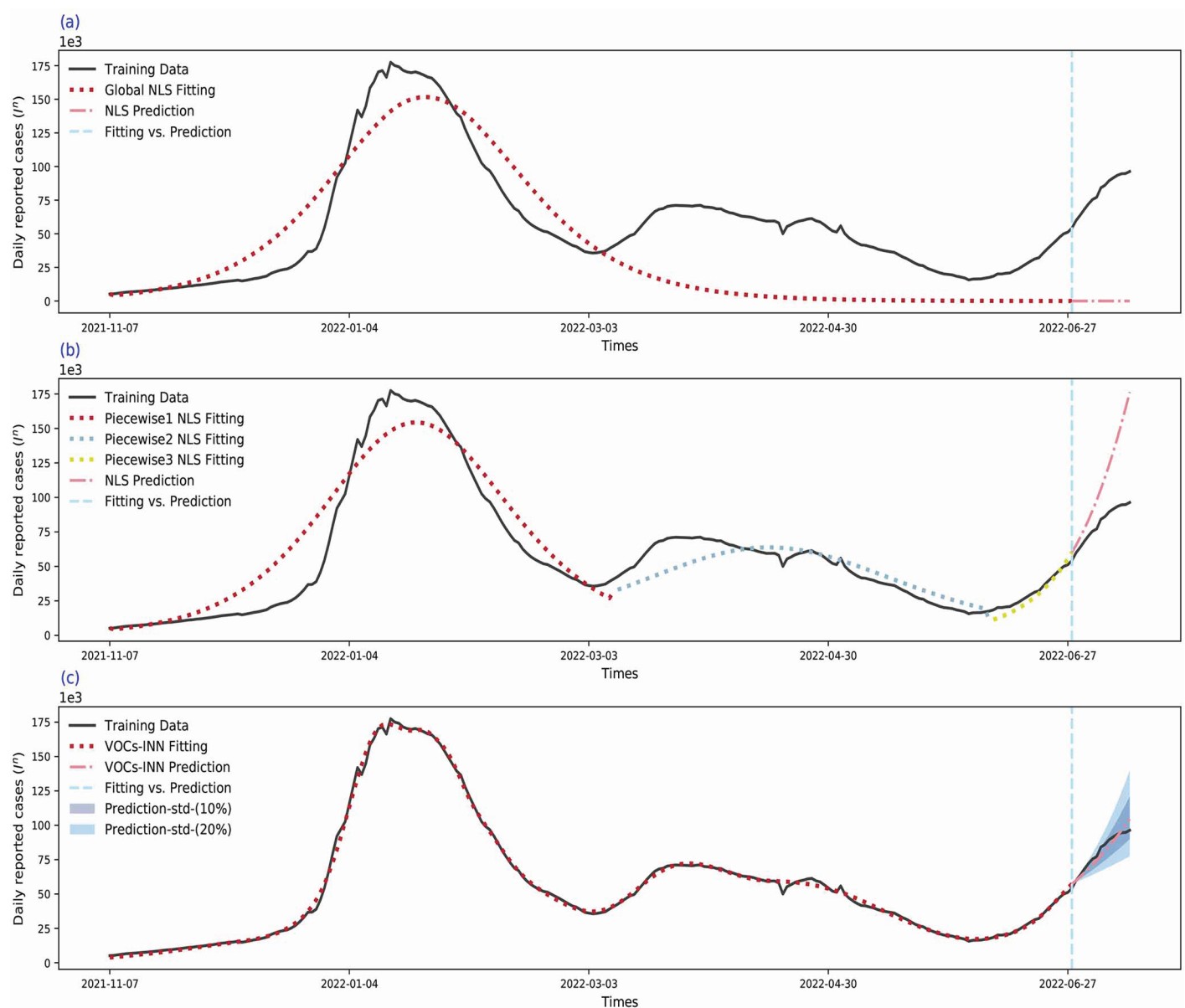

**Fig 12. Comparison of fitting performance and prediction accuracy among three methods.** (a) fitting and prediction using least squares method. (b) Fitting and prediction using piecewise least squares method. (c) Fitting and prediction using VOCs-INN.

**Table 4. Comparison of fitting and prediction performance evaluation indicators.**

| Methods | Least square method | Piecewise least squares method | | VOCs-informed neural network | |
|---|---|---|---|---|---|
| | Fitting | Fitting | Prediction | Fitting | Prediction |
| $R^2$ | 0.4388 | 0.8893 | -7.2195 | 0.9977 | 0.9078 |
| $MSE$ | $1.1274 \times 10^9$ | $2.2236 \times 10^8$ | $1.4931 \times 10^9$ | $4.5858 \times 10^6$ | $1.6743 \times 10^7$ |

the time-varying reproduction number range of the Omicron variant as 1.21–1.95. At the end of 2022, Uriu, Keiya et al. [49] proposed that the effective reproduction number of XBB.1.5 was 1.2 times higher than that of XBB.1, indicating that XBB.1.5 is more infectious. In January

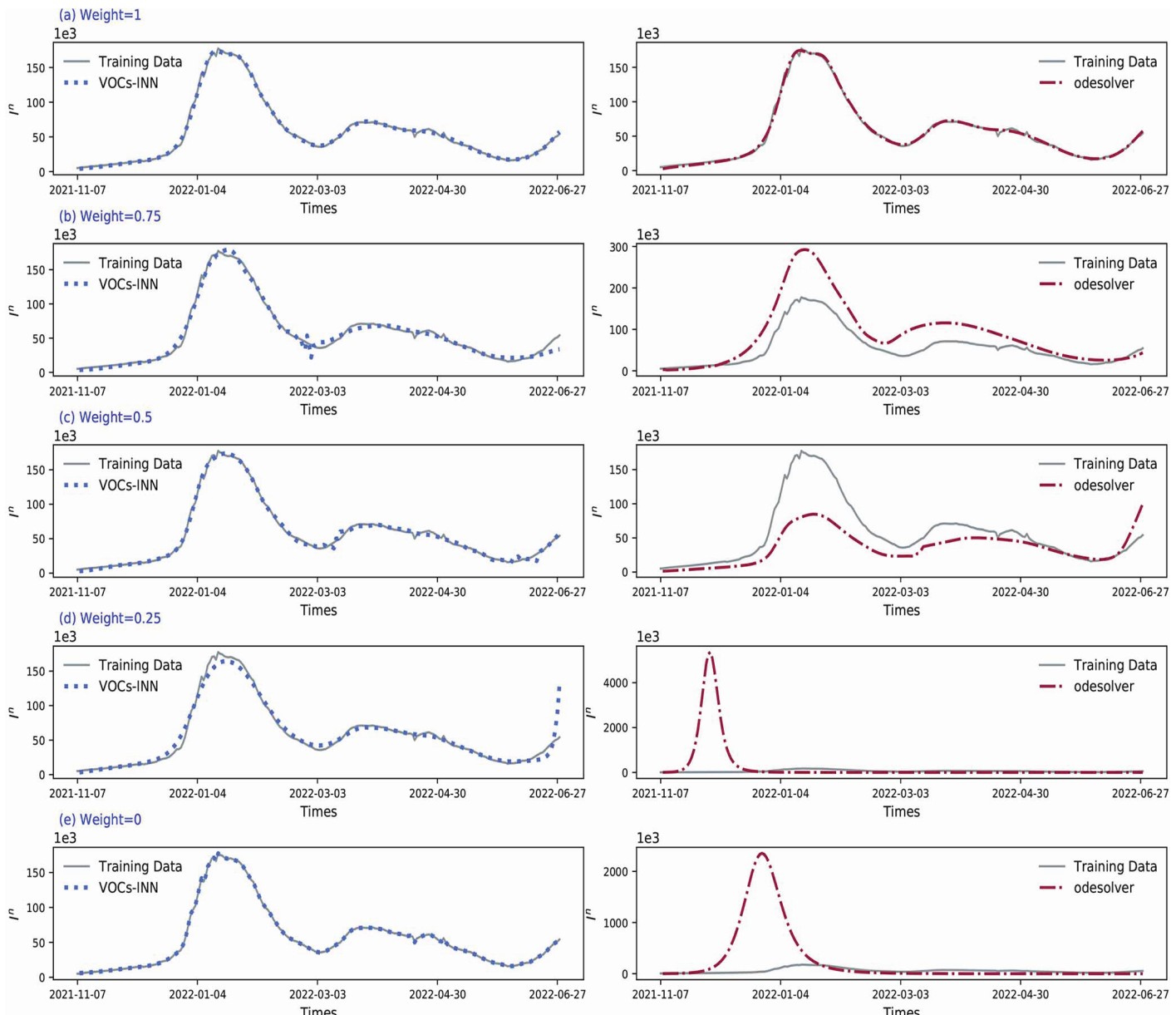

**Fig 13. Comparison of fitting performance and ode solution accuracy with varying weight assignments in the *Loss* function.**

2023, Bloom Lab [50] reviewed previous reports and estimated the effective reproduction number of XBB.1.5 to be around 1.6. By the end of March of that year, Yamasoba, Daichi et al. [51] noted that XBB.1.16 performed better than other variants in India, with an effective reproduction number 1.13 times that of XBB.1.5, which is generally in line with our BC province estimates.

During this period, the estimated values of $\mathcal{R}_{e,1}$ and $\mathcal{R}_{e,2}$ both showed continuous fluctuations and intersected at three different time points. Both values also demonstrated a general downward trend. The observed fluctuations in the effective reproduction numbers ($\mathcal{R}_{e,1}$ and $\mathcal{R}_{e,2}$) can be attributed to two factors. Firstly, during this period, governments took several

intervention measures, such as promoting vaccination, enforcing mandatory mask-wearing in hospitals in April, and lifting the compulsory vaccination requirement for health personnel in July [38–42]. These actions might temporarily reduce $\mathcal{R}_e$, but upon their removal, $\mathcal{R}_e$ could potentially rise again. The observed fluctuations in $\mathcal{R}_e$ may be a result of this cycle of implementing and then lifting these measures. Secondly, in Fig 6(c), we notice that the values of $\mathcal{R}_{e,1}$ initially are higher than those of $\mathcal{R}_{e,2}$. Subsequently, they intersect three times, and ultimately, the values of $\mathcal{R}_{e,2}$ become higher than those of $\mathcal{R}_{e,1}$. These observations suggest that the XBB.1.16* variant initially had a greater presence, then coexisted and competed with the XBB.1.5* variant, which ultimately became more dominant. This aligns with the findings of [52].

Despite the promising results, it is essential to acknowledge the limitations of our study. In the training of neural networks, particularly in regression tasks, the application of a contraction factor is recognized as a fundamental preprocessing step. This step is vital for enhancing numerical stability and improving the overall effectiveness of the model. In our research, we empirically adjusted the contraction factor based on the total population of various regions, aiming to bolster numerical stability. Supporting information illustrates the impact of different contraction factors on the fitting performance and ODE inversion accuracy of our VOCs-INN algorithm. It can be clearly seen from the figure that the results significantly improve when the contraction factor is $10^{-4}$ under this data.

Moreover, the selection of time interpolation points is another crucial factor that significantly influences the model's fitting performance and the accuracy of parameter inference. We meticulously aimed to balance data length and model complexity when determining these points. As depicted in Supporting information, an increase in the number of time interpolation points generally enhances the fitting performance and ODE inversion accuracy of VOCs-INN. However, it is worth noting that this improvement is not linear, excessively large numbers of interpolation points can lead to increased computational time and complexity. Therefore, it is important to note that there is still room for improvement.

Another limitation is that the current multi-strain model does not consider factors such as the latency period, associated fatalities, and possible transitions. Furthermore, while the fixed parameters in the model can be adapted as time-varying parameters using VOCs-INN, this method necessitates a larger and more complex parameter network. These limitations provide avenues for future research to refine further and improve the VOCs-INN algorithm.

In conclusion, the integration of Physics-Informed Neural Networks (PINNs) with infectious disease models, particularly through the proposed VOCs-INN approach, represents a significant research contribution. VOCs-INN leverages available observational data and the underlying physical principles governing infectious disease models to capture the time-varying nature of parameters. This approach enhances the predictive accuracy of the models and provides policymakers with more reliable and scientific frameworks for decision-making. The flexibility of VOCs-INN allows it to be adapted to various epidemiological scenarios and datasets, offering a robust tool for addressing the complexities and uncertainties associated with epidemic challenges. As we continue to neural network into the potential uses of VOCs-INN in infectious disease modeling, we aim to integrate it with other advanced machine-learning methods to better fit and predict infectious disease data.

## Supporting information

**S1 Fig. Weekly confirmed data and weekly cumulative data for the five regions.**
(EPS)

**S2 Fig. Schematic diagram of VOCs-INN for $SI_1I_2R$ model.** $U_D^{NN}(t, \Theta_D)$ is used to fit the state variables of the model (represented by the pink shaded area), while $U_P^{NN}(t, \Theta_P)$ is employed to infer the time-varying parameters (represented by the yellow shaded area). Specifically, $\beta_i^{NN}(t, \Theta_{\beta_i})$ and $\gamma_i^{NN}(t, \Theta_{\gamma_i})$ are used to infer $\beta_i(t)$ and $\gamma_i(t)$ (i=1,2), respectively. $\frac{dU_D^{NN}}{dt}$ denotes the automatic differentiation operator, and the *LOSS* consists of two components, $LOSS_{data}$ and $LOSS_{ode}$. By minimizing the *LOSS*, simultaneous data fitting and inference of time-varying parameters can be achieved.
(EPS)

**S3 Fig. Data fitting and time-dependent parameters inferences for five regions within BC using the VOCs-INN algorithm.** The bar charts display weekly new data and weekly cumulative data. The light blue dashed line represents the best result of the VOCs-INN fitting, while the red dashed line depicts the estimation results of the transmission rates $\beta_1(t)$, $\beta_2(t)$.
(EPS)

**S4 Fig. Comparison of the basic and effective reproduction numbers of two VOIs in five regions.**
(EPS)

**S5 Fig. Prediction of transmission rates $\beta_1$, $\beta_2$ for two VOIs in five regions using the VOCs-INN algorithm.** Blue and red are training and validation data, respectively. The shaded areas indicate the uncertainty ranges obtained by increasing or decreasing the transmission rate by 10% and 20%.
(EPS)

**S6 Fig. Forecasted transmission rates and removal rates utilizing the VOCs-INN approach.** During the model validation and performance assessment, the blue and red dashed lines represent the training data for the VOCs-INN, while the purple dashed line signifies the validation dataset used to simulate actual prediction scenarios. We calculated the predictive range by adjusting $\beta$ and $\gamma$ with an upward and downward fluctuation of 10% and 20%, and presented this range as shaded areas, which gives a probabilistic estimate of the prediction accuracy. The orange vertical dashed line represents the segmentation line between fitting and prediction.
(EPS)

**S7 Fig. Weekly reported cases and cumulative cases are predicted for outbreaks in five regions within BC in 16 weeks from 7 January 2024 (i.e., vertical lines) using a two-VOIs model learned with the VOCs-INN algorithm.**
(EPS)

**S8 Fig. Results of extrapolating the state variable time series for the five regions using VOCs-INN.**
(EPS)

**S9 Fig. Enlarged image of the predicted portion of daily new cases.**
(EPS)

**S10 Fig. Enlarged image of the predicted portion of cumulative new cases.**
(EPS)

**S11 Fig. Parameters inferred from VOCs-INN are used to solve the model using the ODE solver, and the pink dashed lines and blue curves represent the weekly data on new cases reported for the two VOIs in the five regions and the corresponding extrapolated results for the new cases reported each week.**
(EPS)

**S12 Fig. Comparison of fitting performance and ODE solver accuracy of VOCs-INN across different contraction factor.** The gray line represents the actual data, the blue dashed line represents the VOCs-INN fitting line, and the red dashed line represents the ODE solver's reverse calculation line. (a) Contraction factor = 1. (b) Contraction factor = 0.1. (c) Contraction factor = 0.01. (d) Contraction factor = 0.001. (e) Contraction factor = 0.0001.
(EPS)

**S13 Fig. Comparison of fitting performance and ODE solver accuracy of VOCs-INN across time interpolation points.** The gray line represents the actual data, the blue dashed line represents the VOCs-INN fitting line, and the red dashed line represents the ODE solver's reverse calculation line. (a) Time interpolation points = 234. (b) Time interpolation points = 300. (c) Time interpolation points = 500. (d) Time interpolation points = 700. (e) Time interpolation points = 900.
(EPS)

## Author contributions

**Conceptualization:** Wenxuan Li, Xu Chen, Suli Liu, Chiyu Zhang, Guyue Liu.

**Data curation:** Wenxuan Li, Xu Chen.

**Formal analysis:** Wenxuan Li.

**Funding acquisition:** Suli Liu.

**Investigation:** Chiyu Zhang, Guyue Liu.

**Methodology:** Wenxuan Li, Xu Chen, Suli Liu.

**Project administration:** Suli Liu.

**Resources:** Suli Liu.

**Software:** Wenxuan Li, Xu Chen.

**Supervision:** Suli Liu.

**Validation:** Wenxuan Li, Xu Chen, Suli Liu, Chiyu Zhang, Guyue Liu.

**Visualization:** Wenxuan Li, Xu Chen, Suli Liu, Chiyu Zhang, Guyue Liu.

**Writing – original draft:** Wenxuan Li, Xu Chen, Suli Liu.

**Writing – review & editing:** Wenxuan Li, Suli Liu, Chiyu Zhang, Guyue Liu.

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
