## [Decision Letter · Decision Letter 0]

15 Oct 2024

Dear Dr. Liu,

Thank you very much for submitting your manuscript "Using a multi-strain infectious disease model with physical information neural networks to study the time dependence of SARS-CoV-2 variants of concern" for consideration at PLOS Computational Biology.

As with all papers reviewed by the journal, your manuscript was reviewed by members of the editorial board and by several independent reviewers. In light of the reviews (below this email), we would like to invite the resubmission of a significantly-revised version that takes into account the reviewers' comments.

Please also ensure that you are in compliance with our code-sharing policy at this stage.

We cannot make any decision about publication until we have seen the revised manuscript and your response to the reviewers' comments. Your revised manuscript is also likely to be sent to reviewers for further evaluation.

Sincerely,

Roger Dimitri Kouyos

Academic Editor

PLOS Computational Biology

Virginia Pitzer

Section Editor

PLOS Computational Biology

Reviewer's Responses to Questions

**Comments to the Authors:**

Reviewer #1: Based on the limitations in quantifying intervention strategies and using static parameters in SIR models, authors propose a new technique — VOCs-INN — by informing NNs with the mechanisms of SIR models. I find the study very interesting but I have several major points,

1. Given how authors mention the superiority of their method to the previous fitting procedures, I think it is essential to include certain comparisons of simpler approaches in this paper. For example, how would the dummest SIR model would perform with static infection parameters? The next logical step is to compare this with an SIR model that assumes time-dependent variables but does not use NNs for fitting, as well as only NNs what do not have any information from an SIR model. Finally, we reach to the most complex scenario, which is what authors demonstrate with their informed NN model. An easier comparison here can be achieved by playing with the weight of the LOSS_ode relative to LOSS_data and compare the goodness of fit as the weight is increasing from 0 (not informed NN at all) to 1. To make their case, authors should show that informing the NNs with the SIR model (hence increasing this weight) should make significant contributions to the output as it is increasing.

2. This brings me to my second major point, where I couldn’t find any definitions of the goodness of fit in the manuscript. In lines 212-213, authors write :”It can be seen from the figure that VOCs-INN can fit the 212 real data very well. “. This is a computational study, I expect here a mathematical expression that is coherent with the rest of the manuscript. What is the quantification of “fitting well”? Which metric do you use the assess the goodness of fit? The y axis in Fig. 6 is scaled with 1e3, which means very small differences in the plot can map to considerable differences in absolute numbers. A metric to asses the goodness of the fit is necessary here.

3. And the next major point, I think it is also imperative to present the details of computational complexity of these different approaches (for example the dummest SIR versus authors’ method) since this is a trade-off in predictive power versus computational complexity and authors also mention this as a selling point for their method In Lines 21-24. Honestly, I find it hard to believe that fitting NNs is computationally less expensive than a least squares method? I can understand the benefits about the temporal aspect of certain parameters (and most methods assuming them constant over time) but the differences in computational complexity has to be justified if the authors are making this claim.

4. Another major point is about the short-term predictions. Predictions that authors provide in Figs. 9 and 10 correspond to a time where the population is mostly stabilized. Is this really the right time window to demonstrate the capabilities of this model? For example, shouldn’t we test for the short-term predictions of the model right after an intervention that will alter Re? Wouldn’t that be more consistent with the narrative of the paper since it emphasizes this in the authors summary?

5. And finally, reproducibility. I see that there is no repository of the code that is shared? Moreover, hyperparameters and the number of hidden layers is painful to adjust and specific to the question at hand. Authors should mention the processes that led to the results they mention in lines 183-186. What is the learning rate, batch size, etc.? How did the authors experiment with these?

Some of my other minor comments are below.

- I would generally suggest authors to have a more modest language throughout the manuscript, and avoid phrases like “fascinating aspect”.

- As I am at line 72 in the introduction, I am having a hard time understanding what exactly makes this paper novel. Author summary focuses on quantifying the control interventions, then authors mention the inclusion of time dependent parameters, then we read about computational complexity, and then finally incorporating the mechanisms into a black box model and making it more gray-like. What is really the benefit of the method authors developed? All of these? If so, revise writing to make this more clear.

Reviewer #2: Review comments are uploaded as an attachment.

**Have the authors made all data and (if applicable) computational code underlying the findings in their manuscript fully available?**

Reviewer #1: **No: **There is no repository of the code?

Reviewer #2: **No: **No code is shared. The epidemiological data used is named and referenced (ref 35), but the URL in the reference points to example.com. No data underlying the plots is shared. No code or data sharing is discussed.

PLOS authors have the option to publish the peer review history of their article (what does this mean?). If published, this will include your full peer review and any attached files.

Reviewer #1: No

Reviewer #2: No
---

## [Decision Letter · Decision Letter 1]

26 Dec 2024

PCOMPBIOL-D-24-01438R1

Using a multi-strain infectious disease model with physical information neural networks to study the time dependence of SARS-CoV-2 variants of concern

PLOS Computational Biology

Dear Dr. Liu,

Thank you for submitting your manuscript to PLOS Computational Biology. After careful consideration, we feel that it has merit but does not fully meet PLOS Computational Biology's publication criteria as it currently stands. Therefore, we invite you to submit a revised version of the manuscript that addresses the points raised during the review process.

Please submit your revised manuscript within 30 days Feb 25 2025 11:59PM. If you will need more time than this to complete your revisions, please reply to this message or contact the journal office at ploscompbiol@plos.org. Please include the following items when submitting your revised manuscript:

We look forward to receiving your revised manuscript.

Kind regards,

Roger Dimitri Kouyos

Section Editor

PLOS Computational Biology

Roger Kouyos

Section Editor

PLOS Computational Biology

**Journal Requirements:**

1) We have noticed that you have uploaded Supporting Information files, but you have not included a complete list of legends. Please add a full list of legends for your Supporting Information files after the references list.

2) Please ensure that the funders and grant numbers match between the Financial Disclosure field and the Funding Information tab in your submission form. Note that the funders must be provided in the same order in both places as well. State what role the funders took in the study. If the funders had no role in your study, please state: "The funders had no role in study design, data collection and analysis, decision to publish, or preparation of the manuscript.".

**Reviewers' comments:**

Reviewer's Responses to Questions

**Comments to the Authors:**

Reviewer #1: I thank the authors for rigorously addressing all my comments. I believe that the manuscript has improved significantly and ready for publication.

Reviewer #2: Review comments uploaded as an attachment.

**Have the authors made all data and (if applicable) computational code underlying the findings in their manuscript fully available?**

Reviewer #1: Yes

Reviewer #2: Yes

PLOS authors have the option to publish the peer review history of their article (what does this mean?). If published, this will include your full peer review and any attached files.

Reviewer #1: **Yes: **Burcu Tepekule

Reviewer #2: No

**Figure resubmission:**
---

## [Editor Report · Decision Letter 2]

10 Jan 2025

Dear Dr. Liu,

We are pleased to inform you that your manuscript 'Using a multi-strain infectious disease model with physical information neural networks to study the time dependence of SARS-CoV-2 variants of concern' has been provisionally accepted for publication in PLOS Computational Biology.

Best regards,

Roger Dimitri Kouyos

Section Editor

PLOS Computational Biology

Roger Kouyos

Section Editor

PLOS Computational Biology

---

## [Editor Report · Acceptance letter]

PCOMPBIOL-D-24-01438R2

Using a multi-strain infectious disease model with physical information neural networks to study the time dependence of SARS-CoV-2 variants of concern

Dear Dr Liu,

I am pleased to inform you that your manuscript has been formally accepted for publication in PLOS Computational Biology. Your manuscript is now with our production department and you will be notified of the publication date in due course.

With kind regards,

Lilla Horvath
